# Norm-count Hypothesis: On the Relationship Between Norm and Object Count in Visual Representations

## Abstract

We present a novel hypothesis on norms of representations produced by convolutional neural networks (CNNs). In particular, we propose the norm-count hypothesis (NCH), which states that there is a monotonically increasing relationship between the number of certain objects in the image, and the norm of the corresponding representation. We formalize and prove our hypothesis in a controlled setting, showing that the NCH is true for linear and batch normalized CNNs followed by global average pooling, when they are applied to a certain class of images. Further, we present experimental evidence that corroborates our hypothesis for CNN-based representations. Our experiments are conducted with several real-world image datasets, in both supervised, self-supervised, and few-shot learning – providing new insight on the relationship between object counts and representation norms.

## 1 Introduction

The ability to learn high-quality representations from a wide range of complex data types lies at the heart of the success of deep learning. Recently, several works have studied how deep learning-based representations can be embedded in non-Euclidean spaces, to further improve representation quality (Bronstein et al., 2017). In particular, embedding representations on the hypersphere using $L_2$ normalization has proven to be particularly promising direction for several downstream applications, such classification and regression (Mettes et al., 2019; Scott et al., 2021; Tan et al., 2022), self-supervised learning (SSL) (Chen et al., 2020; Caron et al., 2021), and few-shot learning (FSL) (Wang et al., 2019; Fei et al., 2021; Trosten et al., 2023).

However, despite the widespread use of $L_2$ normalization in several aspects of deep learning, little work exists on understanding exactly what type of information the norm contains, and why discarding this information improves representation quality. Hence, we still lack critical understanding of the role of norms and normalization in deep learning. In this work, we aim to improve the understanding of norms of convolutional neural network (CNN)-based representations, and thus to better understand the benefits of $L_2$ normalization. Our work is built on a novel hypothesis for image representations computed by CNNs:

**Informal Definition 1** (Norm-count hypothesis)**.** There is a monotonically increasing relationship between the norm of a representation produced by a CNN, and the number of objects in the image for which the CNN is trained to recognize.

The norm-count hypothesis (NCH) proposes a theory on the relationship between norm, and the number of detections produced by the CNN. Moreover, an implicit consequence of the NCH is that angles encode information about the types of objects detected by the CNN in the given image. In this work, we assess the validity of the NCH for CNNs used in supervised, self-supervised and few-shot learning.

The main contributions of our work are:

1. We propose the NCH – stating that there is a monotonically increasing relationship between the norm of a representation, and the number of objects in the given image.

2. We prove that the NCH is true in a controlled setting, assuming that input images are composed of several *object images*, for which the feature extractor provides a delta-like response in a single channel.

3. We conduct an extensive experimental evaluation with images from MNIST, STL-10, and Pascal VOC, in supervised, self-supervised and few-shot learning. Our results show monotonically increasing relationships between norm and count, for several models and datasets – corroborating the NCH. We also find that discarding norm with $L_2$ normalization improves classification performance in the majority of experimental configurations.

The rest of the paper is structured as follows: Section 2 gives an overview of work related to ours. In Section 3, we theoretically analyze the NCH, and prove that it holds under certain assumptions. Section 4 includes the results of our experiments. We finish the paper with Section 5, presenting some concluding remarks and directions for future work.

## 2 Related work

In this section, we summarize other work related to this paper. We emphasize that our work is complementary to these, as none of the works below provide an accurate and rigorous understanding of the information contained in norms of CNN-based representations.

### 2.1 Embedding representations on the hypersphere

Embedding representations on the hypersphere instead of in Euclidean space has shown to be beneficial for both supervised classification and regression (Mettes et al., 2019; Scott et al., 2021; Tan et al., 2022). Mettes et al. (2019) develop classification and regression losses on the hypersphere, illustrating that $L_2$ normalized representations and prototypes are beneficial for both classification and regression. The more recent work by Tan et al. (2022) shows that a supervised classification model can be regularized with a self-supervised contrastive loss on the unit hypersphere.

$L_2$ normalization is also common in models for self-supervised learning of image representations (Chen et al., 2020; He et al., 2020; Grill et al., 2020; Caron et al., 2020; 2021; Goyal et al., 2022; Li et al., 2023). The benefit of $L_2$ normalization appears to stem from similarity measures and contrastive losses being more well-behaved after discarding the norm – resulting in compact and well-separated classes (Wang & Isola, 2020). However, little work exists on this topic.

Recent methods for transductive FSL have also found $L_2$ normalized representations to be beneficial for classification performance (Wang et al., 2019; Veilleux et al., 2021; Zhu & Koniusz, 2022; Xu et al., 2022; Trosten et al., 2023). Trosten et al. (2023) argue that $L_2$ normalization helps reduce the hubness problem (Radovanovic et al., 2010; Fei et al., 2021) in FSL, and show that embedding points uniformly on the hypersphere completely eliminates hubness. To the best of our knowledge, the work by Trosten et al. (2023) is one of the first works that attempt to understand why $L_2$ normalization is beneficial in FSL. Nevertheless, it is limited to the hubness problem, and does not make any advances in understanding the information contained in the representation norm.

### 2.2 Hyperspherical regularization

Hyperspherical embeddings have also shown to be beneficial to regularize training of deep neural networks (DNNs) (Salimans & Kingma, 2016; Liu et al., 2017; 2018; 2021). These methods constrain the weight vectors in DNNs to lie on the unit hypersphere. Liu et al. (2017) show that hyperspherical weights improve the conditioning of the optimization problem, helping the optimizer converge faster to potentially better solutions. However, our work is orthogonal to this, since we aim to understand norms of *representations*, and not norms of weights.

## 3 Norm-count hypothesis

The purpose of this section is to formalize the NCH, and to analyze it in a rigorous theoretical setting. To do so, we assume certain properties of the feature extractor (*e.g.* CNN). These feature extractors admit a certain class of images, referred to as *object images*, which can be seen as "prototypes" of the objects the feature extractor is trained to detect.

Having established properties of the feature extractor and corresponding object images, we prove that the norm of the representation produced by the feature extractor is proportional to the number of object images present in the given image. Thereby corroborating the NCH. Proofs for the results presented in this section are given in Appendix A.

We start by providing exact definitions of images, image translation, detectors, and global pooling operators.

**Definition 1** (Images). The set of images with $C$ channels and size $W \times H$ is defined as

$$\mathcal{I}_{C,W,H} = \{I : \mathbb{N}^0_{<C} \times \mathbb{Z} \times \mathbb{Z} \to \mathbb{R} \mid I(c,x,y) = 0 \text{ if } (x,y) \notin \mathbb{N}^0_{<W} \times \mathbb{N}^0_{<H}\} \tag{1}$$

where $\mathbb{N}^0_{<a} = \{0, 1, \dots, a-1\}$.

**Definition 2** (Translation operator). A translation operator $\text{Tr}_{x',y'} : \mathcal{I}_{C,W,H} \to \mathcal{I}_{C,W,H}$ shifts the given image by $x', y'$ pixels

$$\text{Tr}_{x',y'}(I)(c,x,y) = I(c, x - x', y - y') \tag{2}$$

**Definition 3** (Translation equivariance). A mapping $f : \mathcal{I}_{C,W,H} \to \mathcal{I}_{C',W',H'}$ is translation equivariant iff for a translation operator $\text{Tr}_{x',y'}$, we have

$$f \circ \text{Tr}_{x',y'} = \text{Tr}_{x',y'} \circ f \tag{3}$$

where $\circ$ denotes the composition of functions.

**Definition 4** (Strict and relaxed detectors). A detector is a *translation equivariant* mapping $f : \mathcal{I}_{C,W,H} \to \mathcal{I}_{C',W',H'}$, which also satisfies at least one of the conditions

$$f(I_1 + I_2) = f(I_1) + f(I_2) + A \tag{4}$$
$$|f(I_1 + I_2)| \preccurlyeq |f(I_1) + f(I_2)| \tag{5}$$

where $A \in \mathcal{I}_{C',W',H'}$ is a constant image independent of $I_1$ and $I_2$, satisfying $A(c,x,y) = a_k$ for all $(x,y) \in \mathbb{N}^0_{<W'} \times \mathbb{N}^0_{<H'}$. Addition and absolute value are defined element-wise, and $I_1 \preccurlyeq I_2$ implies that $I_1(k,x,y) \leq I_2(k,x,y)$ for all $k,x,y$. If $f$ satisfies condition (4), then $f$ is said to be a **strict detector**. If $f$ satisfies condition (5), it is called a **relaxed detector**.

The following propositions show properties of detectors that are relevant for CNNs.

**Proposition 1** (Composition of detectors). *For detectors $f$ and $g$, the following holds:*

  *1. If $f$ and $g$ are strict detectors, then $g \circ f$ is a strict detector.*

  *2. If $f$ is a strict detector and $g$ is a relaxed detector, then $g \circ f$ is a relaxed detector.*

**Proposition 2** (Convolutions are strict detectors.)**.** *Let* $\mathrm{Conv}_K : \mathcal{I}_{C,W,H} \to \mathcal{I}_{1,W+w-1,H+h-1}$ *be the convolution operator convolving the given image,* $I \in \mathcal{I}_{C,W,H}$*, with a filter,* $K \in \mathcal{I}_{C,h,w}$

$$\mathrm{Conv}_K(I)(0, x, y) = \sum_{c=0}^{C-1} \sum_{x'=-\infty}^{\infty} \sum_{y'=-\infty}^{\infty} K(c, x', y') I(c, x - x', y - y'). \tag{6}$$

*Then* $\mathrm{Conv}_K$ *is a strict detector with* $(C', H', W') = (1, W + w - 1, H + h - 1)$*.*

**Proposition 3** (Batch-normalization in inference mode is a strict detector)**.** *Let* $\mathrm{BN}_{\mathbf{b},\mathbf{g},\boldsymbol{\mu},\boldsymbol{\sigma}} : \mathcal{I}_{C,W,H} \to \mathcal{I}_{C,W,H}$ *represent batch normalization in inference mode, with parameters* $\mathbf{b} = [b_0, \ldots, b_{C-1}]^\top, \mathbf{g} = [g_0, \ldots, g_{C'-1}]^\top$ *and running moment estimates* $\boldsymbol{\mu} = [\mu_0, \ldots, \mu_{C-1}]^\top, \boldsymbol{\sigma} = [\sigma_0, \ldots, \sigma_{C'-1}]^\top$*, defined as*

$$\mathrm{BN}_{\mathbf{b},\mathbf{g},\boldsymbol{\mu},\boldsymbol{\sigma}}(I)(k, x, y) = \frac{I(k, x, y) - \mu_k}{\sigma_k} g_k + b_k. \tag{7}$$

*Then* $\mathrm{BN}_{\mathbf{b},\mathbf{g},\boldsymbol{\mu},\boldsymbol{\sigma}}$ *is a strict detector with* $(C', W', H') = (C, W, H)$ *and* $a_k = \frac{\mu_k}{\sigma_k} g_k - b_k$*.*

**Proposition 4** (LeakyReLU is a relaxed detector.)**.** *Let* $\mathrm{LeakyReLU}_\alpha : \mathcal{I}_{C,W,H} \to \mathcal{I}_{C,W,H}$ *be defined element-wise as*

$$\mathrm{LeakyReLU}_\alpha(I(k, x, y)) = \begin{cases} I(k, x, y), & \text{if } I(k, x, y) > 0 \\ \alpha \cdot I(k, x, y), & \text{otherwise} \end{cases} \tag{8}$$

*for all* $k, x, y$*, and* $\alpha \in [0, 1)$*. Then* $\mathrm{LeakyReLU}_\alpha$ *is a relaxed detector with* $(C', W', H') = (C, W, H)$*. This also holds for the standard* $\mathrm{ReLU}(x) = \max\{0, x\}$ *activation, since* $\mathrm{ReLU} = \mathrm{LeakyReLU}_0$*.*

From Propositions 1 and 2, we see that linear and batch normalized CNNs – *i.e.* networks consisting only of compositions of convolutions and batch normalization – are strict detectors. Furthermore, combining Propositions 1, 2 and 4 shows that a CNN consisting of convolutions and LeakyReLU (or ReLU) activations are compositions of relaxed detectors. These propositions thus cover the most important building blocks of CNNs, along with the most common activation functions.

CNNs for classification and representation learning are often followed by a global pooling operator that aggregates information over the spatial dimensions. The following definition considers a general pooling operation, which we use in our theoretical analysis. We then prove that global average pooling (GAP) – one of the most frequently used pooling operations – is a special case of the general pooling operation.

**Definition 5** (Global pooling operator)**.** Let $I \in \mathcal{I}_{C,W,H}$ be an image. The mapping $\mathrm{Pool} : \mathcal{I}_{C,W,H} \to \mathbb{R}^C$ is called a global pooling operator if there exists non-negative real numbers $\gamma_0, \ldots, \gamma_{C-1}$ independent of $I$, such that

$$|\mathrm{Pool}(I)_k| \le \gamma_k \left| \sum_{x=0}^{W-1} \sum_{y=0}^{H-1} I(k, x, y) \right|, \quad k \in \mathbb{N}_{<C}^0. \tag{9}$$

**Proposition 5** (GAP is a global pooling operator)**.** *For an image* $I \in \mathcal{I}_{C,W,H}$*, let GAP be defined as*

$$\mathrm{GAP}(I)_k = \frac{1}{WH} \sum_{x=0}^{W-1} \sum_{y=0}^{H-1} I(k, x, y), \quad k \in \mathbb{N}_{<C}^0. \tag{10}$$

*Then GAP is a global pooling operator with* $\gamma_k = \frac{1}{WH}, \ \forall k \in \mathbb{N}_{<C}^0$*.*

Our objective is now to understand norms of representations computed by a detector followed by a global pooling operator. Hence, we define object images as "prototypical" images that give a delta-like response when processed by the given detector.

**Definition 6** (Object images). An image $O_j \in \mathcal{I}_{C,h,w}$ is said to be an object image of type $j$ w.r.t. the detector $f$, iff

$$f(O_j) = \delta_{j,0,0} \tag{11}$$

where $\delta_{j,x',y'}$ is the Kronecker delta function

$$\delta_{j,x',y'}(k,x,y) = \begin{cases} 1, & (k,x,y) = (j,x',y') \\ 0, & \text{otherwise} \end{cases}. \tag{12}$$

The set of all object images of type $j$ is denoted $\Omega_j = \{O \mid f(O) = \delta_{j,0,0}\}$.

In a supervised classification setting, the object image types will tend to coincide with the classes the model is trained to detect. This is because supervised models are trained to output a one-hot prediction vector, resembling the delta-response assumed in Definition 6, after global pooling.

Object images can also be interpreted as "parts of a whole", where it is assumed that image motifs consist of a collection of object images. An image of a car, for instance, will be composed of object images with type "wheel", "car body", *etc.*

We note that, by definition, object images and types are entirely determined by the detector, as images that give a delta response in a single output feature map. All images that result in a delta response in an output feature map, $j$, are said to be object images of type $j$. Hence, it is only for trained models that the object image types coincide with semantically meaningful ground-truth classes.

We will now define multiple objects images (MO-images) as images composed of one or more object images. This definition gives rise to a natural notion of object "count" in the image, which is necessary to formalize the NCH.

**Definition 7** (Multiple objects image). An MO-image, $I \in \mathcal{I}_{C,W,H}$, constructed from object images in $\Omega_0, \ldots, \Omega_{C'-1}$ is defined as

$$I = \sum_{j=0}^{C'-1} \sum_{(O,x',y') \in \mathcal{P}_j} T_{(x',y')}(O). \tag{13}$$

where $C'$ is the number of object types. $\mathcal{P}_j$ is a set of 3-tuples, where the first element is an object image from $\Omega_j$, and the second third elements are the positions of that object image in $I$.

We now have the following theorem stating that the NCH is true for detectors and global pooling operators applied to MO-images.

**Theorem 1** (Norm-count hypothesis – simplified setting). *Let $f : \mathcal{I}_{C,W,H} \to \mathcal{I}_{C',W',H'}$ be a relaxed detector with object images $\Omega_0, \ldots, \Omega_{C'-1}$, and let $I \in \mathcal{I}_{C,W,H}$ be a MO-image constructed from the same object images. Then, if $\mathbf{z} = [z_0, \ldots, z_{C'-1}]^\top \in \mathbb{R}^{C'}$ is the output of a global pooling operator applied to the feature maps $f(I)$, we have*

$$|z_k| = |\operatorname{Pool}(I)_k| \leq \gamma_k |\mathcal{P}_k|, \quad k \in \mathbb{N}^0_{<C'} \tag{14}$$

*for non-negative numbers $\gamma_0, \ldots, \gamma_{C-1}$ independent of $I$.*

**Corollary 1.1** ($L_p$ norm of $\mathbf{z}$). *For $p > 0$, the $L_p$ norm of $\mathbf{z}$ from Theorem 1 is*

$$||\mathbf{z}||_p \leq \left( \sum_{k=0}^{C'-1} (\gamma_k |\mathcal{P}_k|)^p \right)^{\frac{1}{p}} \tag{15}$$

Corollary 1.1 shows that the $L_p$ norm of representations is upper bounded by a monotonically increasing function of the count, corroborating the NCH.

**Corollary 1.2** (Strict detector and GAP)**.** *If $f$ is a strict detector, and* GAP *is used in place of* Pool, *Theorem 1 simplifies to*

$$z_k = \text{GAP}(I)_k = \frac{|\mathcal{P}_k|}{W'H'} + n_{\mathcal{P}} a_k \tag{16}$$

*where $n_{\mathcal{P}} = \sum_{j=0}^{C'-1} |\mathcal{P}_j| - 1$.*

Corollary 1.2 shows that for strict detectors followed by GAP, there is an affine relationship between each component of $\mathbf{z}$, and the number of objects of the corresponding type present in the image. Furthermore, if we have $a_k = 0$, we have exact proportionality between $z_k$ and $|\mathcal{P}_k|$. Since linear CNNs are strict detectors with $a_k = 0$, this corollary proves that each dimension, in a representation produced by linear CNNs, is proportional to the number of object images in the given MO-image.

Furthermore, Theorem 1 states that the norm of $\mathbf{z}$ is directly related to the absolute (total) count of objects in the image, regardless of the type of the object images. This is expected, since the perfect detector produces a set of delta-responses, and the global pooling operator aggregates these over the spatial dimensions.

In contrast to the norm, the angle of $\mathbf{z}$ depends on the count of one object type relative to the count of another object type. This is demonstrated by the following result.

**Result 1** (Semantic information in angles)**.** *Suppose $I_1, I_2 \in \mathcal{I}_{C,W,H}$ are MO-images processed by a strict detector $f$ with $a_k = 0$, followed by GAP. Furthermore, assume that $I_1$ only consists of objects of type $j$, and that $I_2$ only consists of objects of type $k$. This gives*

$$\mathbf{z}_1 = \text{GAP}(f(I_1)) = \frac{|\mathcal{P}_j^{(1)}|}{W'H'}\mathbf{e}_j \quad and \quad \mathbf{z}_2 = \text{GAP}(f(I_2)) = \frac{|\mathcal{P}_k^{(2)}|}{W'H'}\mathbf{e}_k \tag{17}$$

*where $\mathbf{e}_j$ ($\mathbf{e}_k$) denotes the vector where element $j$ ($k$) is $1$, and all other elements are $0$.*
*Then, if $(||\mathbf{z}||, \boldsymbol{\theta}(\mathbf{z}))$ denotes the transformation of $\mathbf{z}$ to hyperspherical coordinates, we can consider the following two cases:*

1. Different class, same count*: $j \neq k$ and $|\mathcal{P}_j^{(1)}| = |\mathcal{P}_k^{(2)}|$, which gives*

$$||\mathbf{z}_1|| = ||\mathbf{z}_2|| \quad and \quad \boldsymbol{\theta}(\mathbf{z}_1) \neq \boldsymbol{\theta}(\mathbf{z}_2) \tag{18}$$

2. Same class, different count: *$j = k$ and $|\mathcal{P}_j^{(1)}| \neq |\mathcal{P}_k^{(2)}|$, which gives*

$$||\mathbf{z}_1|| \neq ||\mathbf{z}_2|| \quad and \quad \boldsymbol{\theta}(\mathbf{z}_1) = \boldsymbol{\theta}(\mathbf{z}_2) \tag{19}$$

In both cases in Result 1, the angles $\boldsymbol{\theta}(\mathbf{z}_1)$ and $\boldsymbol{\theta}(\mathbf{z}_2)$ are most informative of the image classes (object types). When the images belong to different classes (case 1), the discriminative power lies in the angles and not in the norms. Conversely, when $I_1$ and $I_2$ belong to the same class (case 2), the within class distance is 0 for the angles, but non-zero for the norms. The angle thus encodes information about which classes (object types) that were detected in the image – *i.e.* the semantic information.

**Feature norm and object size.** CNNs are not size equivariant, meaning that the size of an object will not necessarily be positively correlated with the representation norm. This is because convolutions – the basic building blocks of CNNs – detect patterns with a certain size. Resizing the patterns by contracting or dilating spatial dimensions can therefore completely change the response, both reducing og increasing its strength. In the context of our work, this means that object images are not resizeable: If one resizes an object image, it may no longer be an object image for the same detector, as it might no longer give a delta response. This, however depends on the properties of the detector, and whether it has been trained to

produce similar responses for objects with different sizes. Appendix B includes an experiment with objects of increasing size, showing a *negative* correspondence between object size and feature norm for a supervised model.

## 4 Experiments

The purpose of these experiments is to experimentally investigate the NCH in a controlled setting. We design the experiments to have fine-grained control of the "count" in each image. This allows us to properly examine the relationship between norm and count – both quantitatively and qualitatively. Our experiments are conducted with both supervised, self-supervised and few-shot learning models.

Although our theoretical results hold for arbitrary $L_p$ norms, we focus on $L_2$ norms in the experimental evaluation. This is because the $L_2$ norm is the one most frequently encountered in other works (see Section 2), and has known benefits related to optimization (Liu et al., 2021), as well as alignment, uniformity, and class separability (Wang & Isola, 2020).

### 4.1 Setup

**Models and architectures.** Our evaluation is performed with models using the following two CNN architectures:

- `Simple-6`: A simple 6-layer CNN followed by GAP. The model has ReLU activations, and max pooling after every second convolutional layer. No batch-normalization or other forms of normalization is applied anywhere in the architecture.

- `ResNet-50`: The standard 50-layer residual network architecture by He et al. (2016), with batch normalization. We use the supervised model available in Torchvision[1] For the self-supervised version of this architecture, we use the model available in Lightning Bolts[2].

In Appendix C we include additional experiments with Dense Contrastive Learning (Wang et al., 2021) – a self-supervised model aimed to provide better representations for multi-label data.

**Training details.** All models are trained on single images, using a cross-entropy loss for the supervised models, and the standard contrastive loss for the SimCLR models. The `Simple-6` models were trained on MNIST by us. For the `ResNet-50` models we downloaded pre-trained versions from the respective sources. None of the model configuration use $L_2$ normalization on the representations during training (although SimCLR applies $L_2$ normalization to the *projections* produced by the projection head).

**Datasets.** In order to mimic the properties of MO-images in evaluation, we start with datasets consisting of natural images (MNIST (Lecun et al., 1998) and STL-10 (Coates et al., 2011)). Then, to generate a single evaluation image, we sample a random number of images, and place them at random positions in a $4 \times 4$ grid. This gives us an image that resembles an MO-image, where we know the true count – *i.e.* the number of object images.

These grid-based datasets are generated either with all images in the grid from the same class (single label), or with random class affiliations (mixed) for the small images.

For MNIST, we fill the empty grid positions with 0-values, as indicated by Definition 7. For STL-10, we generate datasets with 3 different approaches to filling the empty grid slots:

- *Zeros*: empty grid positions are filled with 0-values.

- *Random*: empty grid positions are filled with random Gaussian noise with the same mean and standard deviation as the object images.

---

[1] https://pytorch.org/vision/stable/models/resnet.html
[2] https://lightning-bolts.readthedocs.io/en/0.3.4/self_supervised_models.html.

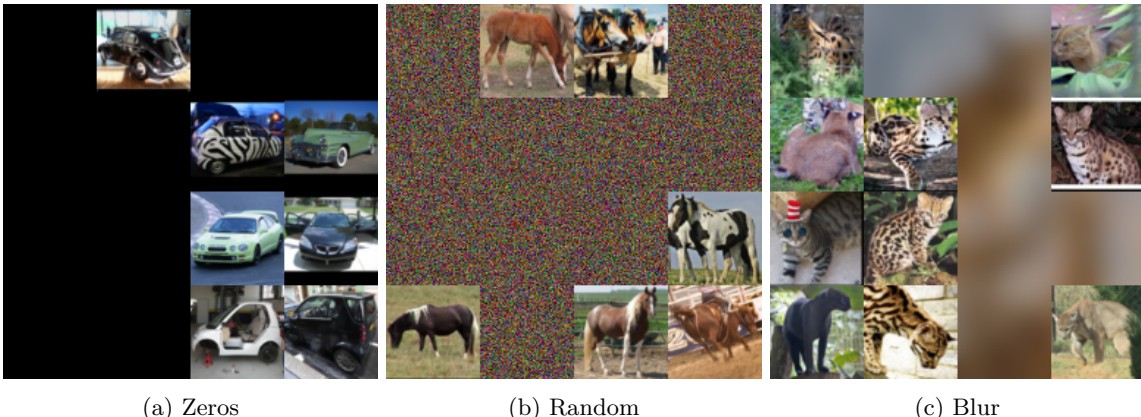

(a) Zeros             (b) Random             (c) Blur

Figure 1: Example images generated from STL-10, using the different filling approaches.

- *Blur*: the background for the whole grid is a random blurred image, and the object images are placed on top of this image.

We experiment with different fill modes to ensure that the results are not skewed by changes in global image statistics, such as mean and variance. Figure 1 shows examples of the generated images.

In addition to the grid datasets, we use the Pascal VOC 2007–2013 and COCO 2014 object counting/detection datasets (Everingham et al., 2010; Lin et al., 2014) to evaluate the relationship between norm and count in real images. The object count in an image is computed as the total number of ground-truth bounding boxes in the image. We then select images with $2 \leq \text{count} \leq 10$.

We note that the VOC and COCO datasets can contain objects that are not labeled, meaning that they do not have a bounding box, and are thus not included in the count. This makes VOC and COCO more challenging, but also more realistic, compared to the grid datasets.

**Quantitative evaluation of monotonic increase.** We use a weighted quadratic regression model to quantitatively assess whether there is an increasing relationship between the feature norm, $||\mathbf{z}_i||$, and the count, $c(\mathbf{x}_i)$

$$||\mathbf{z}_i|| = \beta_0 + \beta_1 c(\mathbf{x}_i) + \beta_2 c(\mathbf{x}_i)^2 + \epsilon_i \tag{20}$$

where the residual, $\epsilon_i$, is assumed to be Gaussian with zero mean and standard deviation $\sigma_{c(\mathbf{x}_i)}$. We allow the standard deviation to be count-dependent to account for heteroskedasticity in the data. Further, we use a quadratic regressor to have a more robust evaluation of monotonicity between norm and count. As can be seen in Figure 2, the relationship between norm and count is not always linear. A quadratic regressor fits the data better, giving a more precise measure of monotonicity.

The parameter estimates $(\hat{\beta}_0, \hat{\beta}_1, \hat{\beta}_2)$ are computed using weighted least squares. Based on these estimates, we can test for monotonic increase by checking whether the derivative

$$\frac{\mathrm{d}\,||\mathbf{z}||}{\mathrm{d}\,c(\mathbf{x})} = \hat{\beta}_1 + 2\,\hat{\beta}_2\,c(\mathbf{x}) \tag{21}$$

is positive.

We report the value of the slope $s_q = \hat{\beta}_1 + 2\,\hat{\beta}_2\,c_q$, and the *p*-value resulting from testing

$$H_0 : \hat{\beta}_1 + 2\,\hat{\beta}_2\,c_q \leq 0 \quad \text{vs.} \quad H_1 : \hat{\beta}_1 + 2\,\hat{\beta}_2\,c_q > 0. \tag{22}$$

The *p*-value is denoted by $p_q$.

Table 1: Estimated slopes ($s_q$) and $p$-values ($p_q$) for the quadratic regression model for norm vs. count. A positive slope with a low $p$-value indicates a statistically significant, monotonically increasing relationship between norm and count.

| Dataset | Model | Fill | Labels | $s_{0.25}$ | $s_{0.5}$ | $s_{0.75}$ | $p_{0.25}$ | $p_{0.5}$ | $p_{0.75}$ |
|---|---|---|---|---|---|---|---|---|---|
| MNIST | SimCLR (`Simple-6`) | Zeros | Mixed | 0.147 | 0.206 | 0.264 | 0.000 | 0.000 | 0.000 |
| | | | Single | 0.131 | 0.190 | 0.249 | 0.000 | 0.000 | 0.000 |
| | Supervised (`Simple-6`) | Zeros | Mixed | -0.075 | 0.142 | 0.359 | 1.000 | 0.000 | 0.000 |
| | | | Single | 0.027 | 0.152 | 0.277 | 0.000 | 0.000 | 0.000 |
| STL-10 | SimCLR (`ResNet-50`) | Blur | Mixed | 0.288 | 0.203 | 0.117 | 0.000 | 0.000 | 0.000 |
| | | | Single | 0.295 | 0.202 | 0.109 | 0.000 | 0.000 | 0.000 |
| | | Random | Mixed | 0.253 | 0.194 | 0.136 | 0.000 | 0.000 | 0.000 |
| | | | Single | 0.260 | 0.196 | 0.132 | 0.000 | 0.000 | 0.000 |
| | | Zeros | Mixed | 0.231 | 0.207 | 0.183 | 0.000 | 0.000 | 0.000 |
| | | | Single | 0.237 | 0.210 | 0.183 | 0.000 | 0.000 | 0.000 |
| | Supervised (`ResNet-50`) | Blur | Mixed | 0.196 | 0.132 | 0.069 | 0.000 | 0.000 | 0.000 |
| | | | Single | 0.143 | 0.073 | 0.003 | 0.000 | 0.000 | 0.404 |
| | | Random | Mixed | 0.292 | 0.152 | 0.012 | 0.000 | 0.000 | 0.052 |
| | | | Single | 0.250 | 0.105 | -0.039 | 0.000 | 0.000 | 1.000 |
| | | Zeros | Mixed | 0.305 | 0.184 | 0.062 | 0.000 | 0.000 | 0.000 |
| | | | Single | 0.283 | 0.144 | 0.004 | 0.000 | 0.000 | 0.317 |
| VOC | SimCLR (`ResNet-50`) | – | Mixed | -0.017 | -0.017 | -0.016 | 1.000 | 1.000 | 0.996 |
| | | | Single | 0.001 | 0.016 | 0.030 | 0.391 | 0.051 | 0.019 |
| | Supervised (`ResNet-50`) | – | Mixed | 0.081 | 0.059 | 0.037 | 0.000 | 0.000 | 0.000 |
| | | | Single | 0.065 | 0.047 | 0.030 | 0.000 | 0.000 | 0.024 |
| COCO | SimCLR (`ResNet-50`) | – | Mixed | 0.029 | 0.006 | -0.017 | 0.000 | 0.088 | 0.990 |
| | | | Single | 0.012 | -0.006 | -0.024 | 0.000 | 0.794 | 0.982 |
| | Supervised (`ResNet-50`) | – | Mixed | 0.065 | 0.037 | 0.008 | 0.000 | 0.000 | 0.133 |
| | | | Single | 0.043 | 0.034 | 0.025 | 0.000 | 0.000 | 0.024 |

The slopes and $p$-values are computed at equally spaced points, $c_{0.25}$, $c_{0.5}$, and $c_{0.75}$, where

$$c_q = c_{\min} + q \cdot (c_{\max} - c_{\min}) \tag{23}$$

and $c_{\min}$ and $c_{\max}$ are the minimum and maximum counts in the dataset, respectively.

**Few-shot learning experiments.** In our few-shot experiments, we use the `ResNet-50` models described above, and evaluate on STL-10 and VOC, where we know the ground-truth counts. We report both 1 and 5-shot classification accuracy, using the Simpleshot classifier (Wang et al., 2019). The evaluation includes 10000 episodes, sampling 5 random classes and 15 queries from each class, in each episode.

**Implementation.** The experiments are implemented in Python with the PyTorch framework (Paszke et al., 2019). Our implementation will be made publicly available upon publication of the paper.

## 4.2 Results: supervised and self-supervised learning

**Relationship between norm and count.** Table 1 lists the results of the regression analyses for norm vs. count for the different configurations. The slopes and $p$-values show that the majority of configurations

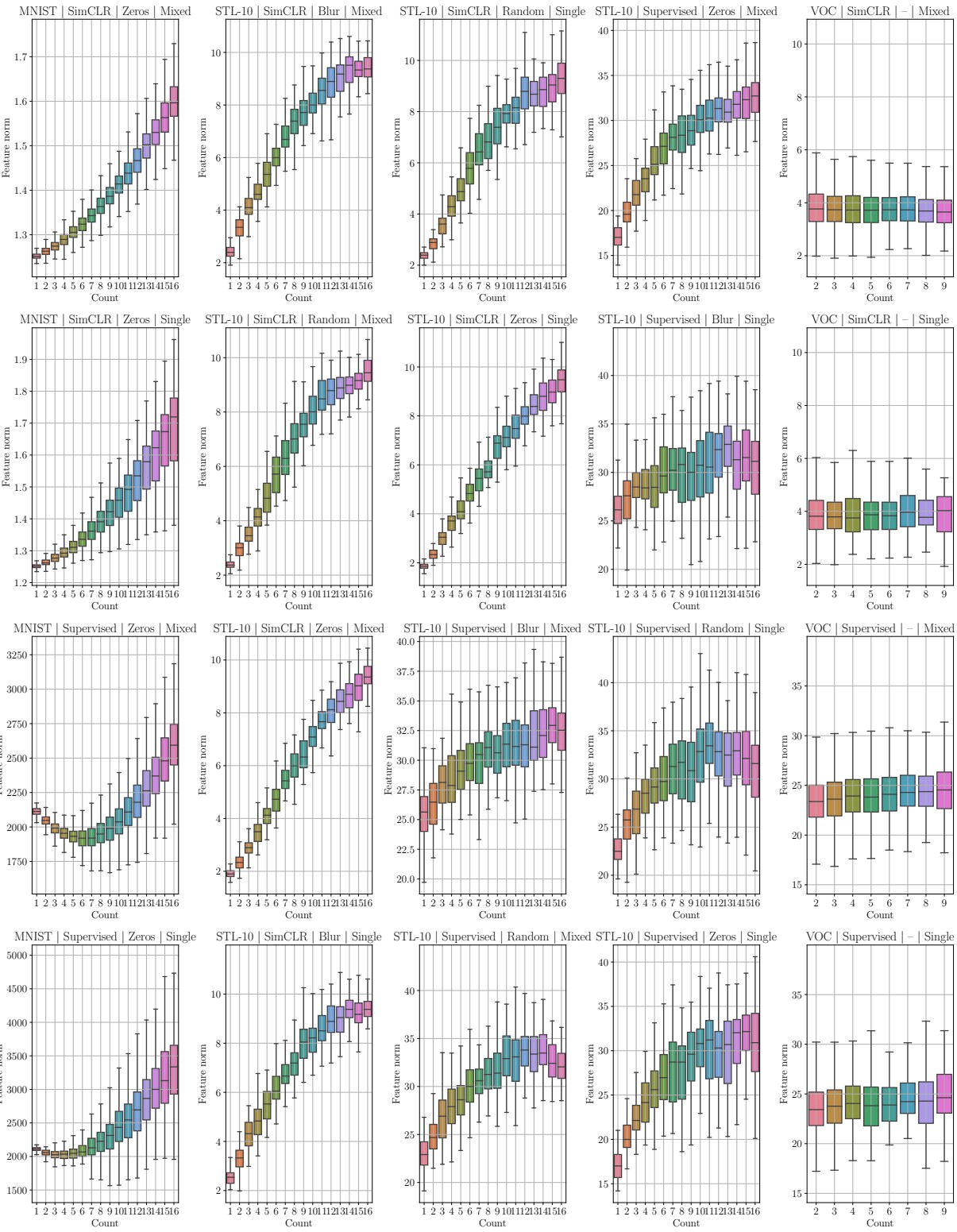

Figure 2: Boxplots illustrating the relationship between norm and count for the experimental configurations.

Table 2: Differences in classification accuracy [%] after $L_2$ normalizing the representations, for the prototypical and SGD classifiers. Positive values indicate an that $L_2$ normalization improves classification accuracy.

| Dataset | Model | Fill | $\Delta$ (Proto) | $\Delta$ (SGD) |
|---|---|---|---|---|
| MNIST | SimCLR (`Simple-6`) | Zeros | 2.92 | 0.0 |
| | Supervised (`Simple-6`) | Zeros | 3.78 | -1.01 |
| STL-10 | SimCLR (`ResNet-50`) | Blur | 13.4 | 0.2 |
| | | Random | 10.8 | -0.7 |
| | | Zeros | 11.4 | 0.4 |
| | Supervised (`ResNet-50`) | Blur | 1.9 | 0.6 |
| | | Random | 1.2 | 0.2 |
| | | Zeros | 5.8 | 1.1 |

| Dataset | Model | $\Delta$ (Proto) | $\Delta$ (SGD) |
|---|---|---|---|
| VOC | SimCLR (`ResNet-50`) | 2.55 | 1.4 |
| | Supervised (`ResNet-50`) | -0.49 | 1.44 |
| COCO | SimCLR (`ResNet-50`) | 3.98 | -3.06 |
| | Supervised (`ResNet-50`) | -0.06 | -4.99 |

result in a statistically significant, monotonically increasing relationship between norm and count. The same trend can be observed in the box-plots in Figure 2. This applies to both the grid-based datasets (MNIST and STL-10), and most configurations ran on natural images (VOC and COCO).

However, for the supervised model on STL-10 and VOC, we observe that the increasing trend is not as clear as for the other models. This is likely because the supervised `ResNet-50` is trained on the ImageNet dataset. This dataset contains natural images with a varying number of objects. Since the model is trained to output *the same prediction regardless of the number of objects*, the model learns to be approximately *count invariant*, resulting in a weaker relationship between norm and count.

Finally, we observe no increasing trend between norm and count for some SimCLR runs on VOC and COCO. These are the most challenging configurations, since the images depict complex scenes with multiple objects of different type, and the model is trained without supervision. We hypothesize that the non-increasing relationship between norm and count is a result of SimCLR detecting objects that do not necessarily coincide with the labels. The object images for this model do therefore not correspond to the ground-truth objects, resulting in a norm that is no longer representative of the count.

$L_2$ **normalization and classification performance.** Table 2 shows the change in classification accuracy after $L_2$ normalizing the representations. The accuracies are computed both based on a prototypical classifier (classifying samples according to the closest class mean), and a linear classifier trained with stochastic gradient descent (SGD). These results show that $L_2$ normalization – *i.e.* discarding the norm – is in most cases beneficial for classification performance, resulting in increased accuracy. The increased accuracy after normalization illustrates that the norm contains little relevant class information, and acts as additional noise in the classifier. These findings are in line with the theory in Result 1, stating that angles carry most of the semantic information. In some cases however, we observe a drop in accuracy after $L_2$ normalization. This is likely because the model has learned to encode other types of information in the norm, thereby violating the delta assumption of the NCH – especially for the `ResNet-50` modes trained on ImageNet, where the training images might contain several objects.

### 4.3 Results: few-shot learning

Table 3 shows 1 and 5-shot classification results on the STL-10, VOC and COCO datasets. We observe that $L_2$ normalization results in a statistically significant improvement in performance for all models and datasets – illustrating that discarding the norm information is beneficial for few-shot classification performance.

We also compute the average correlation between norm and count ($\rho_{\text{norm,count}}$) across episodes. The correlation values show that also for few samples, there is a clear correlation between norm and count. This holds for all configurations, except for VOC with the SimCLR model. Similar to in Section 4.2, we hypothesize

Table 3: 1 and 5-shot few-shot learning results on STL-10 and VOC, with the Simpleshot classifier (Wang et al., 2019). The results are averaged across 10000 episodes, with 95 % confidence intervals shown in parentheses. The correlation between norm and count, $\rho_{\mathrm{norm,count}}$, could only be computed for evaluations without $L_2$ normalization, since the norm has zero variance after $L_2$ normalization.

| Dataset | Model | Fill | Norm | 1-shot Accuracy | $\rho_{\mathrm{norm,count}}$ | 5-shot Accuracy | $\rho_{\mathrm{norm,count}}$ |
|---|---|---|---|---|---|---|---|
| STL-10 | SimCLR (`ResNet-50`) | Blur | None | 0.431 (0.006) | 0.95 (0.001) | 0.56 (0.006) | 0.952 (0.001) |
| | | | $L_2$ | 0.522 (0.007) | − (−) | 0.655 (0.006) | − (−) |
| | | Random | None | 0.432 (0.005) | 0.95 (0.001) | 0.558 (0.006) | 0.952 (0.001) |
| | | | $L_2$ | 0.514 (0.007) | − (−) | 0.654 (0.006) | − (−) |
| | | Zeros | None | 0.437 (0.006) | 0.95 (0.001) | 0.56 (0.006) | 0.952 (0.001) |
| | | | $L_2$ | 0.518 (0.006) | − (−) | 0.654 (0.006) | − (−) |
| | Supervised (`ResNet-50`) | Blur | None | 0.774 (0.006) | 0.323 (0.008) | 0.882 (0.005) | 0.155 (0.012) |
| | | | $L_2$ | 0.801 (0.006) | − (−) | 0.925 (0.003) | − (−) |
| | | Random | None | 0.754 (0.006) | 0.491 (0.007) | 0.894 (0.003) | 0.504 (0.007) |
| | | | $L_2$ | 0.772 (0.007) | − (−) | 0.919 (0.003) | − (−) |
| | | Zeros | None | 0.717 (0.006) | 0.674 (0.005) | 0.872 (0.003) | 0.676 (0.005) |
| | | | $L_2$ | 0.786 (0.007) | − (−) | 0.914 (0.003) | − (−) |
| VOC | SimCLR (`ResNet-50`) | − | None | 0.511 (0.007) | -0.109 (0.008) | 0.721 (0.006) | -0.116 (0.006) |
| | | | $L_2$ | 0.651 (0.006) | − (−) | 0.765 (0.006) | − (−) |
| | Supervised (`ResNet-50`) | − | None | 0.688 (0.007) | 0.044 (0.007) | 0.826 (0.005) | 0.086 (0.006) |
| | | | $L_2$ | 0.742 (0.006) | − (−) | 0.827 (0.006) | − (−) |
| COCO | SimCLR (`ResNet-50`) | − | None | 0.457 (0.011) | 0.045 (0.006) | 0.767 (0.008) | 0.087 (0.005) |
| | | | $L_2$ | 0.568 (0.009) | − (−) | 0.818 (0.006) | − (−) |
| | Supervised (`ResNet-50`) | − | None | 0.586 (0.01) | 0.184 (0.007) | 0.841 (0.006) | 0.118 (0.006) |
| | | | $L_2$ | 0.625 (0.01) | − (−) | 0.842 (0.006) | − (−) |

that this is because SimCLR detects another set of object images, whose counts are not well-represented by the ground-truth object count.

## 5  Conclusion and future work

We have presented the NCH – providing novel insight in the norms of representations produced by CNNs. Under certain assumptions on the model and input images, we proved the NCH, showing that each component in a given representation is upper-bounded by the number of object images present in the input image. Moreover, from our theoretical analysis, it follows that representation norms carry information related to count, whereas angles represent semantic information. This is a key result that helps explain why $L_2$ normalization is beneficial for downstream classification tasks.

In addition, we conduct a controlled experimental evaluation, showing that the NCH holds for supervised and self-supervised models. Our experiments also show that discarding the representation norm – which is strongly correlated with the count – using $L_2$ normalization, improves classification performance, both for standard and few-shot classifiers.

We believe that understanding representation norms through object counts is a promising direction of research. The popularity of hyperspherical embeddings in FSL, along with our results on the benefits of $L_2$ normalization, indicate that there is extra potential to improve FSL models if we better understand the effects of representation norms. Furthermore, although we focus on CNNs in this work, it is entirely possible that our results will generalize to other architectures. This can be verified by either showing that certain architectures meet the detector conditions, or by relaxing these conditions and extending the theoretical analysis.

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

# A Proofs

## A.1 Proposition 1

*Proof.*

1. Invoking condition (4) of strict detectors gives

$$(g \circ f)(I_1 + I_2) = g(f(I_1 + I_2) = g(f(I_1) + f(I_2)) \tag{24}$$
$$= g(f(I_1)) + g(f(I_2))) = (g \circ f)(I_1) + (g \circ f)(I_2) \tag{25}$$

2. By conditions (4) and (5), we have

$$|(g \circ f)(I_1 + I_2)| = |g(f(I_1 + I_2) = g(f(I_1) + f(I_2))| \tag{26}$$
$$\leq |g(f(I_1)) + g(f(I_2)))| = |(g \circ f)(I_1) + (g \circ f)(I_2)| \tag{27}$$

$\square$

## A.2 Proposition 2

*Proof.* The proposition follows directly from convolutions being linear and translation equivariant. See *e.g.* (Jähne, 2002, Ch. 4). $\square$

## A.3 Proposition 3

*Proof.* For images $I_1$ and $I_2$ we have

$$\mathrm{BN}_{\mathbf{b},\mathbf{g},\boldsymbol{\mu},\boldsymbol{\sigma}}(I_1 + I_2)(k,x,y) = \frac{I_1(k,x,y) + I_2(k,x,y) - \mu_k}{\sigma_k} g_k + b_k \tag{28}$$

$$= \frac{g_k}{\sigma_k}(I_1(k,x,y) + I_2(k,x,y)) + b_k - \frac{\mu_g g_k}{\sigma_k} \tag{29}$$

$$= \left( \frac{g_k}{\sigma_k} I_1(k,x,y) + b_k - \frac{\mu_g g_k}{\sigma_k} \right) + \left( \frac{g_k}{\sigma_k} I_2(k,x,y) + b_k - \frac{\mu_g g_k}{\sigma_k} \right) - \left( b_k - \frac{\mu_k g_k}{\sigma_k} \right) \tag{30}$$

$$= \mathrm{BN}_{\mathbf{b},\mathbf{g},\boldsymbol{\mu},\boldsymbol{\sigma}}(I_1)(k,x,y) + \mathrm{BN}_{\mathbf{b},\mathbf{g},\boldsymbol{\mu},\boldsymbol{\sigma}}(I_2)(k,x,y) + \left( \frac{\mu_k g_k}{\sigma_k} - b_k \right) \tag{31}$$

$\square$

## A.4 Proposition 4

*Proof.* Let $a_1 = I_1(k,x,y)$ and $a_2 = I_2(k,x,y)$. Since addition is commutative, we can assume $a_1 \geq a_2$ without loss of generality.

Observe that, if $a_1$ and $a_2$ is positive (negative), then $a_1 + a_2$ will be positive (negative). This means that $\mathrm{LeakyReLU}_\alpha(a_1 + a_2) = \mathrm{LeakyReLU}_\alpha(a_1) + \mathrm{LeakyReLU}_\alpha(a_2)$ in this case.

On the other hand, if $a_2 < 0 < a_1$, we have $|\mathrm{LeakyReLU}_\alpha(a_1) + \mathrm{LeakyReLU}_\alpha(a_2)| = |a_1 + \alpha a_2|$, and

$$|\mathrm{LeakyReLU}_\alpha(a_1 + a_2)| = \begin{cases} |a_1 + a_2|, & |a_1| > |a_2| \\ \alpha|a_1 + a_2|, & \text{otherwise} \end{cases}. \tag{32}$$

Since $\alpha \in [0,1)$, we have

$$|\mathrm{LeakyReLU}_\alpha(a_1 + a_2)| \leq |a_1 + a_2| \leq |a_1 + \alpha a_2| = |\mathrm{LeakyReLU}_\alpha(a_1) + \mathrm{LeakyReLU}_\alpha(a_2)| \tag{33}$$

$\square$

### A.5 Proposition 5

*Proof.* Setting $\frac{1}{WH} = \gamma_k$ gives

$$GAP(I)_k = \gamma_k \sum_{x=0}^{W-1} \sum_{y=0}^{H-1} I(k, x, y), \quad k \in \mathbb{N}^0_{<C} \tag{34}$$

from which it follows that

$$|GAP(I)_k| \leq \gamma_k \left| \sum_{x=0}^{W-1} \sum_{y=0}^{H-1} I(k, x, y) \right|, \quad k \in \mathbb{N}^0_{<C} \tag{35}$$

$\square$

### A.6 Theorem 1

*Proof.* Since $f$ is a relaxed detector, we have

$$|f(I)| = \left| f \left( \sum_{j=0}^{C'-1} \sum_{(O,x',y') \in \mathcal{P}_j} T_{(x',y')}(O) \right) \right| \preccurlyeq \left| \sum_{j=0}^{C'-1} \sum_{(O,x',y') \in \mathcal{P}_j} f(T_{(x',y')}(O)) \right| \tag{36}$$

Then, since $f$ is translation equivariant, and provides delta detections

$$|f(I)| \preccurlyeq \left| \sum_{j=0}^{C'-1} \sum_{(O,x',y') \in \mathcal{P}_j} f(T_{(x',y')}(O)) \right| = \left| \sum_{j=0}^{C'-1} \sum_{(i,x',y') \in \mathcal{P}_j} \delta_{(j,x',y')} \right|. \tag{37}$$

Applying a global pooling operator to $f(I)$ then gives

$$|z_k| = |\operatorname{Pool}(f(I))_k| \leq \gamma_k \left| \sum_{x=0}^{W'-1} \sum_{y=0}^{H'-1} f(I)(k, x, y) \right| \tag{38}$$

$$\leq \gamma_k \left| \sum_{x=0}^{W'-1} \sum_{y=0}^{H'-1} \left( \sum_{j=0}^{C'-1} \sum_{(i,x',y') \in \mathcal{P}_j} \delta_{(j,x',y')}(k, x, y) \right) \right| \tag{39}$$

$$\leq \gamma_k \left| \sum_{x=0}^{W'-1} \sum_{y=0}^{H'-1} \sum_{(i,x',y') \in \mathcal{P}_k} \delta_{(k,x',y')}(k, x, y) \right| \tag{40}$$

$$\leq \gamma_k \left| \sum_{(i,x',y') \in \mathcal{P}_k} \delta_{(k,x',y')}(k, x', y') \right| \tag{41}$$

$$\leq \gamma_k |\mathcal{P}_k| \tag{42}$$

$\square$

### A.7 Corollary 1.1

*Proof.* The $L_p$ norm of $\mathbf{z}$ is defined as

$$||\mathbf{z}||_p = \left( \sum_{k=0}^{C'-1} |z_k|^p \right)^{\frac{1}{p}} \tag{43}$$

for $p > 0$. Since each $|z_k|$ is positive and upper bounded by $\gamma_k |\mathcal{P}_k|$ (by Theorem 1), we have

$$\sum_{k=0}^{C'-1} \gamma_k |\mathcal{P}_k|^p \geq \sum_{k=0}^{C'-1} |z_k|^p \tag{44}$$

which gives

$$\left( \sum_{k=0}^{C'-1} \gamma_k |\mathcal{P}_k|^p \right)^{\frac{1}{p}} \geq \left( \sum_{k=0}^{C'-1} |z_k|^p \right)^{\frac{1}{p}} = ||\mathbf{z}||_p \tag{45}$$

$\square$

## A.8   Corollary 1.2

*Proof.* This proof follows the same steps as the proof of Theorem 1, but without absolute values, and with equality instead of inequality.

Since $f$ is a strict detector, we have

$$f(I) = f \left( \sum_{j=0}^{C'-1} \sum_{(O,x',y') \in \mathcal{P}_j} T_{(x',y')}(O) \right) \tag{46}$$

$$= \sum_{j=0}^{C'-1} \sum_{(O,x',y') \in \mathcal{P}_j} \left( f(T_{(x',y')}(O)) + T_{(x',y')}(A) \right) \tag{47}$$

$$= n_{\mathcal{P}} A + \sum_{j=0}^{C'-1} \sum_{(O,x',y') \in \mathcal{P}_j} f(T_{(x',y')}(O)) \tag{48}$$

where $n_{\mathcal{P}} = \sum_{j=0}^{C'-1} |\mathcal{P}_j| - 1$. Then, since $f$ is translation equivariant, and provides delta detections

$$f(I) = n_{\mathcal{P}} A + \sum_{j=0}^{C'-1} \sum_{(O,x',y') \in \mathcal{P}_j} f(T_{(x',y')}(O)) = n_{\mathcal{P}} A + \sum_{j=0}^{C'-1} \sum_{(i,x',y') \in \mathcal{P}_j} \delta_{(j,x',y')}. \tag{49}$$

Applying a global pooling operator to $f(I)$ then gives

$$z_k = \text{GAP}(f(I))_k = \frac{1}{W'H'} \sum_{x=0}^{W'-1} \sum_{y=0}^{H'-1} f(I)(k,x,y) \tag{50}$$

$$= \frac{1}{W'H'} \sum_{x=0}^{W'-1} \sum_{y=0}^{H'-1} \left( n_{\mathcal{P}} A(k,x,y) + \sum_{j=0}^{C'-1} \sum_{(O,x',y') \in \mathcal{P}_j} \delta_{(j,x',y')}(k,x,y) \right) \tag{51}$$

$$= \frac{1}{W'H'} \sum_{x=0}^{W'-1} \sum_{y=0}^{H'-1} \left( n_{\mathcal{P}} a_k + \sum_{(O,x',y') \in \mathcal{P}_k} \delta_{(k,x',y')}(k,x,y) \right) \tag{52}$$

$$= \frac{1}{W'H'} \sum_{(i,x',y') \in \mathcal{P}_k} \delta_{(k,x',y')}(k,x',y') + n_{\mathcal{P}} a_k \tag{53}$$

$$= \frac{1}{W'H'} |\mathcal{P}_k| + n_{\mathcal{P}} a_k \tag{54}$$

$\square$

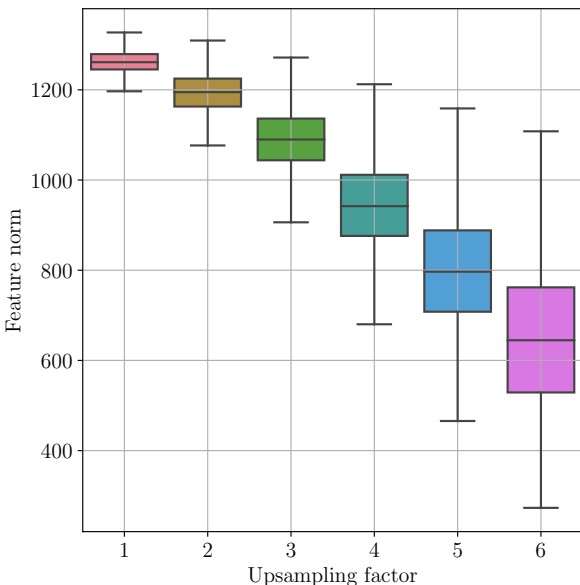

Figure 3: Distributions of representation norms for MNIST digits with different upsampling factors.

Table 4: Experiments with Dense Contrastive Learning.

| Dataset | Model | Fill | Labels | $s_{0.25}$ | $s_{0.5}$ | $s_{0.75}$ | $p_{0.25}$ | $p_{0.5}$ | $p_{0.75}$ |
|---------|-------|------|--------|-----------|-----------|-----------|-----------|-----------|-----------|
| STL-10 | DenseCL (`ResNet-50`) | Blur | Mixed | 0.257 | 0.204 | 0.152 | 0.000 | 0.000 | 0.000 |
| | | | Single | 0.237 | 0.191 | 0.144 | 0.000 | 0.000 | 0.000 |
| | | Random | Mixed | 0.190 | 0.200 | 0.210 | 0.000 | 0.000 | 0.000 |
| | | | Single | 0.181 | 0.195 | 0.209 | 0.000 | 0.000 | 0.000 |
| | | Zeros | Mixed | 0.174 | 0.207 | 0.240 | 0.000 | 0.000 | 0.000 |
| | | | Single | 0.171 | 0.202 | 0.232 | 0.000 | 0.000 | 0.000 |
| VOC | DenseCL (`ResNet-50`) | – | Mixed | -0.022 | -0.014 | -0.006 | 1.000 | 1.000 | 0.839 |
| | | | Single | -0.000 | -0.002 | -0.004 | 0.518 | 0.590 | 0.612 |
| COCO | DenseCL (`ResNet-50`) | – | Mixed | 0.006 | -0.004 | -0.014 | 0.001 | 0.804 | 0.970 |
| | | | Single | 0.023 | -0.008 | -0.039 | 0.000 | 0.852 | 0.999 |

## B    Experiments with varying object size

Figure 3 shows distributions of representation norms of MNIST digits with varying upsampling factors. The images are resized to a random integer factor in $\{1, \ldots, 6\}$, and placed in an empty ($168 \times 168$) image. The plot shows a *negative* correspondence between size and feature norm, meaning that object size and count influence the feature norm in different ways.

## C    Experiments with Dense Contrastive Learning

The results in Table 4 shows that there is a monotonically increasing relationship between norm and count on STL-10, and on low counts on COCO. On VOC and higher counts on COCO we do not observe the same trend. This is likely caused by the ImageNet models being approximately count invariant, as described in Section 4.

