# OpenReview forum: "Norm-count Hypothesis: On the Relationship Between Norm and Object Count in Visual Representations"
_TMLR — Rejected by TMLR_

### Review · Reviewer_tSBH · 2023-09-15

**Summary Of Contributions:**

This paper introduces the 'norm-count hypothesis,' which suggests a positive correlation between the number of objects in an image and the norm of the feature. Both theoretical and empirical results are provided to validate this hypothesis.

**Audience:**

Yes

**Broader Impact Concerns:**

no problem with the ethical implications of the work

**Claims And Evidence:**

No

**Requested Changes:**

Please see the above comment. The rating is major revision.

**Strengths And Weaknesses:**

[Strengths]

+ The paper is easy to follow.

[ Weaknesses & Suggestions to Authors]

- The core content of the hypothesis is somewhat problematic. The hypothesis posits that "an increase in the number of objects in an image leads to a greater norm of the network feature." This hypothesis mainly centers on the count of the objects, which does not take the object size into consideration. However, the size of an object obviously impacts the feature norm. As the size of an object increases, the number of activated units in the intermediate features may correspondingly increase. Thus, the omission of object size in the hypothesis casts some uncertainty on the completeness of it.

-The theoretical findings might be problematic. The hypothesis is supposed to be applied to well-trained networks. However, in the theoretical derivation section, the paper does not make any assumptions regarding the network's classification capabilities or related properties, which merely assumes the neural network employs convolution and leaky ReLU operations. One might question whether the theory derived also pertains to a network with completely randomized initialization.
- The definition of object categories in the paper appears somewhat ambiguous. The author does not provide a clear definition of what constitutes an "object category." Should objects within the same category have similar their appearance? If objects of the same category have different appearances and sizes, e.g., two chairs with fully different appearances, then the hypothesis will not be valid.

- In the experimental section, the authors did not clarify the rationale behind employing a quadratic function to validate the norm-count hypothesis. Why not design a regression model based on the equation in Corollary 1.1 (e.g., designing a regression model with a mathematical form similar to Equation (14)) to solve the counting problem. The choice of a quadratic fit appears somewhat ad-hoc.

- The experimental framework could be enhanced by exploring the influence of object sizes on the feature norm.

- Does the author's theory pertain universally to every layer within the network? I suggest that the author use features of different layers to test the relationship between the object number and the feature norm.

---

> ### Author Response · Authors · 2023-10-18
> **Reply to Reviewer tSBH**
>
> We thank the reviewer for their feedback, and are glad that they found our paper to be easy to follow. Our replies to the reviewer's comments are listed below.
>
> 1. Convolutional neural networks (CNNs) are not size equivariant, meaning that the size of an object will not necessarily be positively correlated with the representation norm. This is because convolutions -- the basic building blocks of CNNs -- detect patterns with a certain size. Resizing the patterns by contracting or dilating spatial dimensions can therefore completely change the response, both reducing og increasing its strength. In the context of our work, this means that object images are not resizeable: If one resizes an object image, it may no longer be an object image for the same detector, as it might no longer give a delta response. This, however depends on the properties of the detector, and whether it has been trained to produce similar responses for objects with different sizes. We have updated the paper with a brief discussion on object sizes, and we have included an experiment with varying size in Appendix B. This experiment shows that the relationship between norm and count is different from the relationship between norm and size.
>
> 2. The delta assumption made for object images is closely related to the model's classification performance. In particular, the object image types (resulting in delta responses) will tend to coincide with ground-truth classes when the model is able to discriminate between classes with high accuracy. In our experiments we include both supervised (trained to classify ground-truth classes) and self-supervised models (trained with contrastive learning). Object images (providing delta responses in the output feature maps) might also exist for randomly initialized networks, but these will most likely appear random and semantically meaningless.
>
> 4. Object categories are entirely defined by the detector as images that give a delta response in a single output feature map. All images that result in a delta response in an output feature map, _j_, are said to be object images of type _j_. As mentioned above, it is only for trained models that the object image types will coincide with semantically meaningful ground-truth classes. We have updated the paper with a more thorough description of object images and object image types.
>
> 5. We used a quadratic regressor to have a more robust evaluation of monotonicity between norm and count. As can be seen in Figure 2, the relationship between norm and count is not always linear. A quadratic function fits the data better, giving a more precise measure of monotonicity. We have updated the manuscript with a brief justification of this choice.
>
> 6. We have updated the paper with a brief discussion on object sizes, and we have included an experiment with varying size in Appendix B.
>
> 7. The NCH applies to all models that meet the relevant assumptions: relaxed/strict detector followed by a global pooling operator. If the NCH holds for a certain network, it will also hold for a truncated version of the same network, as long as a global pooling operator is applied. However, the object images of the truncated network would likely be different from the object images associated with the full network.

---

> > ### Comment · Reviewer_tSBH · 2023-10-25
> > **commnets**
> >
> > I thank the authors for providing some additional clarifications, which helps me understand the paper better. However, some concerns still have not been fully solved in the authors' responses.
> >
> > The experiment about varing object size is not sufficient. The author only conducted experiments on simple datasets like MNIST. Could the author evaluate the relationship between object size and feature norm on some other realistic dataset through object segment annotations, such as the COCO dataset or any other larger benchmark datasets?

---

> > > ### Author Response · Authors · 2023-11-06
> > > **Answer to comments**
> > >
> > > We thank the Reviewer for the continued interest in our work and acknowledge the Reviewer's concerns about the experiments and arguments regarding object size. However, we have already included the following paragraph in the manuscript explaining why object size is not expected to have the same impact on feature norm, compared to object count:
> > >
> > > >CNNs are not size equivariant, meaning that the size of an object will not necessarily be positively correlated with the representation norm. This is because convolutions -- the basic building blocks of CNNs -- detect patterns with a certain size. Resizing the patterns by contracting or dilating spatial dimensions can therefore completely change the response, both reducing and increasing its strength. In the context of our work, this means that object images are not resizeable: If one resizes an object image, it may no longer be an object image for the same detector, as it might no longer give a delta response. This, however depends on the properties of the detector, and whether it has been trained to produce similar responses for objects with different sizes. Appendix B includes an experiment with objects of increasing size, showing a _negative_ correspondence between object size and feature norm for a supervised model.
> > >
> > > Although the experiments in Appendix B are conducted on MNIST, they still present a valid counterexample, illustrating that the relationship between feature norm and object size is not monotonically increasing.
> > >
> > > If the Reviewer has additional insight suggesting that feature norm and object size have a monotonically increasing relationship, we kindly ask them to present experimental or theoretical evidence supporting this claim. In that case we are happy to provide additional experimental and/or theoretical results to further clarify the relationship between norm and size.

---

### Review · Reviewer_xgij · 2023-09-18

**Summary Of Contributions:**

The paper presents the Norm-Count Hypothesis (NCH), which states that the norm of a representation produced by a CNN is positively correlated with the number of objects in the image. The paper demonstrates that such a hypothesis is true for networks composed of convolutions and LeakyReLU layers, when appropriately defining what is a detector and an object image (i.e. an image/patch containing an object that triggers a detector). Then it conducts empirical experiments on different datasets with two different architectures (a linear CNN with 6 layers and ResNet-50, trained with SimCLR). These experiments show to some extent that the NCH also holds for these models, and that L2-normalizing the representations of those networks helps in obtaining better classification accuracy.

**Audience:**

Yes

**Broader Impact Concerns:**

No concerns.

**Claims And Evidence:**

No

**Requested Changes:**

* Show whether the NCH could hold for networks with normalization layers.

* Test the hypothesis with natural images.

* Directly test the hypothesis on a counting task using the norm of a representation.

* Understand and explain why discarding the norm would help classification if knowing the number of objects in an image could be benefitial.

**Strengths And Weaknesses:**

Strengths:

* The NCH is a good insight that holds, as defined in the paper, for linear CNNs.
* The authors conducted a large amount of experiments.

Weaknesses:

* The NCH is only demonstrated for linear networks, without normalization. All current vision models (CNNs, ViTs) use normalization and therefore such hypothesis doesn't hold since most representations have unit norm. Therefore, the applicability of the NCH is reduced to older, less performant models.

* The datasets used to test the hypothesis are synthetic - they are collages of images from different datasets. This allows control of the  number of objects in the collage images, but these images are not natural. Instead, the authors could have used images from object detection datasets where the number of objects in the image is well established and the images are natural.

* If the hypothesis was true, it would imply that we can compare the number of objects in multiple images by comparing the norm. If so, then the norm could be used to solve object counting tasks. It would be useful to then directly test the hypothesis by using the norms from the representations of a standard model to propose a solution for a counting task. Example of counting task: TallyQA, Acharya et al. AAAI 2019

* The paper shows empirically that L2-normalizing representations increases classification accuracy. However, the argument presented is not very convincing. Paraphrasing my understading of the argument - since the norm encodes information about the number of objects, this is not directly useful for classification, so when removed the classification accuracy is better. I have two issues with this reasoning: first, I am not sure that knowing the number of objects in an image is not useful for classification. Second, it seems unlikely that discarding useless information would increase classification accuracy. In general, I'm not sure how proposing NCH is related to L2-normalization improving classification accuracy, and I would like the argument to be more formally made in the paper.

---

> ### Author Response · Authors · 2023-10-18
> **Reply to Reviewer xgij**
>
> We thank the reviewer for their feedback, and appreciate that they found the NCH to be sound. Our replies to the reviewer's comments are listed below.
>
> 1. We have extended our theoretical analysis to include batch normalized networks. Specifically, we have updated the definition of strict detectors, and included a proposition showing that batch normalization in inference is a strict detector. We have also updated Corollary 1.2 accordingly.
>
> 2. The experiments already include an object detection dataset, namely Pascal VOC. This dataset contains natural images with a known number of objects. The results on this dataset are presented alongside the results on the synthetic datasets in Figure 2 and in Tables 1-3.
>
> 3. We have included additional experiments on the MS-COCO object counting/detection dataset, which is one of the datasets from the list suggested by reviewer zszo. This dataset contains natural images with a varying number of objects, similar to VOC. See the revised manuscript for results on the COCO dataset.
>
> 4. Most classification tasks disregard the number of objects in the image: e.g. an image should be classified as a certain class regardless of how many instances there are of that class in the image. In this case the classification task is _count invariant_, meaning that count acts as noise (irrelevant information) in the classification task. We therefore expect that reducing count information through normalization will make the representations less noisy, thereby improving classification performance. However, in cases where count _is_ relevant to the classification task, discarding the norm (and thereby the count) with normalization, would result in worse performance.

---

> > ### Comment · Reviewer_xgij · 2023-11-06
> >
> > I thank the authors for their comments and the implemented changes. I am still not convinced with the new changes, for the following reasons:
> >
> > - The theoretical part of the paper does take into consideration things like network initialization or the performance of the trained network, or ignores the relation between object size and feature norm, as mentioned by other reviewers.
> > - In real world datasets, it seems the hypothesis does not hold empirically. The correlation of norm and count for Pascal VOC and COCO is really low. Figure 2 last column shows this visually, where the mean of the representation norm for images of different object counts is basically the same.
> > - I still don't fully agree that being count-invariant is a good property for an image classification system. Furthermore, in the experiment the network is trained without L2-normalization, which means it is optimized to solve the task while being count-variant according to the hypothesis in the paper. Why not train the network directly with L2-normalization and show that it can have the same or better performance?

---

> > > ### Author Response · Authors · 2023-11-06
> > >
> > > We thank the reviewer for their continued interest in our work. We will attempt to address the Reviewer's concerns in the following.
> > >
> > > - It is true that the theoretical contributions do not explicitly depend on initialization. This is because the hypothesis holds, in principle, for any CNN regardless of initialization -- _presuming that object images exist for that network_. For a randomly initialized network, object images might not exist, or appear random and meaningless.  However, assuming that object images coincide with ground truth objects (meaning that ground-truth objects provide delta responses in the detector), implicitly assumes that the network has been trained to high accuracy. The theoretical contributions thus takes network initialization into account through the delta assumption.
> > >
> > >   Please also see these replies to Reviewer tSBH in the rebuttal:
> > >   >2.  The delta assumption made for object images is closely related to the model's classification performance. In particular, the object image types (resulting in delta responses) will tend to coincide with ground-truth classes when the model is able to discriminate between classes with high accuracy. In our experiments we include both supervised (trained to classify ground-truth classes) and self-supervised models (trained with contrastive learning). Object images (providing delta responses in the output feature maps) might also exist for randomly initialized networks, but these will most likely appear random and semantically meaningless.
> > >   >3.  Object categories are entirely defined by the detector as images that give a delta response in a single output feature map. All images that result in a delta response in an output feature map, _j_, are said to be object images of type _j_. As mentioned above, it is only for trained models that the object image types will coincide with semantically meaningful ground-truth classes. We have updated the paper with a more thorough description of object images and object image types.
> > >
> > >   Regarding object size, please see this answer to Reviewer tSBH:
> > >   >[...] we have already included the following paragraph in the manuscript explaining why object size is not expected to have the same impact on feature norm, compared to object count:
> > >   >
> > >   >_CNNs are not size equivariant, meaning that the size of an object will not necessarily be positively correlated with the representation norm. This is because convolutions -- the basic building blocks of CNNs -- detect patterns with a certain size. Resizing the patterns by contracting or dilating spatial dimensions can therefore completely change the response, both reducing and increasing its strength. In the context of our work, this means that object images are not resizeable: If one resizes an object image, it may no longer be an object image for the same detector, as it might no longer give a delta response. This, however depends on the properties of the detector, and whether it has been trained to produce similar responses for objects with different sizes. Appendix B includes an experiment with objects of increasing size, showing a _negative_ correspondence between object size and feature norm for a supervised model._
> > >   >
> > >   >Although the experiments in Appendix B are conducted on MNIST, they still present a valid counterexample, illustrating that the relationship between feature norm and object size is not monotonically increasing.
> > >   >
> > >   >If the Reviewer has additional insight suggesting that feature norm and object size have a monotonically increasing relationship, we kindly ask them to present experimental or theoretical evidence supporting this claim. In that case we are happy to provide additional experimental and/or theoretical results to further clarify the relationship between norm and size.
> > >
> > > - The correlation is low in Pascal VOC, indicating that there are other factors contributing to the norm, in addition to object count. However, the quantitative results still show a _statistically significant_ increasing relationship, corroborating the NCH.
> > > - Count invariance is a desirable property of the model _if the task it is designed to solve is count invariant_. Many (but not all) classification tasks are inherently count invariant: An image containing a number of objects of type X should be classified as X, regardless of the count.
> > >
> > >   In the experiments we trained our models without L2 normalization to match standard classification/self-supervised models as best as possible. These experiments corroborate the NCH despite the lack of L2 normalization. The positive effects of L2 normalization on classification performance have already been documented in several related works. See e.g. the following references in the manuscript: Mettes et al. 2019, Tan et al. 2022, Chen et al. 2020, Caron et al. 2020, Li et al. 2023, Wang & Isola 2020,  Wang et al. 2019, Zhu & Koniusz 2022, Trosten et al. 2023, Radovanovic et al. 2010.

---

### Review · Reviewer_zszo · 2023-09-24

**Summary Of Contributions:**

- The paper presents an analysis on the norm of a learned representation and the number of objects in the image.
- The paper finds that there is a monotonic increasing relationship between norm of a representation and the # objects. Apart from empirical validation, this is theoretically proved under some assumptions.
- I am not aware of any such similar analysis in literature.

**Audience:**

Yes

**Broader Impact Concerns:**

The paper does not have a broader impact section but I don't think it is strictly necessary for this paper.

**Claims And Evidence:**

No

**Requested Changes:**

Please refer to the weaknesses. In addition, following are some requested changes:

1. Theoretical analysis : Eq 14 : it would be good to give the readers some intuition on what the bounds might convey and how tight they are. It would also be helpful to list down all assumptions for the theoretical analysis at a single place.
2. How strong is the assumption on delta-response for the datasets considered ?
3. The authors can also consider testing with DINO-based SSL models which have been shown to have good semantic features and object understanding capabilities out of the box.

**Strengths And Weaknesses:**

Strengths:
- The paper is well written and easy to follow for the most part.
- Findings of the paper are interesting, especially the ones in real-world empirical setups.
- The final draft of the paper might of benefit to the wider community and result in further analysis on this topic.


Weaknesses/Questions:
- The synthetic data generation approach introduces lot of artifacts not common in natural images : abrupt boundaries, noise etc which might affect the conclusions. While the authors have shown experiments on datasets like VOC, I think that instead of the synthetic dataset it would be better to include more of such real-world datasets.
- It will be interesting to see if the results are correlated with the quality of representations : by reporting the performance of the learned representations on downstream task like classification.
- The paper does not show any results on some real-world counting datasets. I am not an expert on this topic but a quick search leads to the following benchmarks[c].
- How were the models with MO datasets trained ? Multi-label training, would this affect the assumptions of the theoretical analysis ?
- Prior work has shown that vanilla SimCLR is not the best in dealing with self-supervised pre-training on multi-label datasets. Use of features from papers like [a,b] might lead to different conclusions.
- Experiments with L2-normalization : How are these experiments carried out ? Are the features from the extractor L2 normalized during training ?


[a] Dense Contrastive Learning for Self-Supervised Visual Pre-Training, CVPR-2021
[b] Object-aware Cropping for Self-Supervised Learning, TMLR-2022
[c] https://paperswithcode.com/task/object-counting

---

> ### Author Response · Authors · 2023-10-18
> **Reply to Reviewer zszo**
>
> We thank the reviewer for their feedback. Further, we appreciate that they found our paper interesting, and found our findings to be relevant for a broader audience. Our replies to the reviewer's comments are listed below.
>
> 1. We have included additional experiments on the MS-COCO object counting/detection dataset, which is one of the datasets from the list suggested by the reviewer. This dataset contains natural images with a varying number of objects, similar to VOC. See the revised manuscript for results on the COCO dataset.
>
> 2. Table 2 shows changes in downstream classification accuracy before and after L2 normalization. In most cases, we observe that L2 normalizing representations (and thereby discarding count information) leads to better classification performance. Better representations wrt. downstream classification accuracy will imply that the delta-assumption is more likely to be satisfied, with the object images corresponding to ground-truth classes. This will result in a better correspondence between norm and object count, meaning that discarding norm aligns better with discarding object count.
>
> 3. We have included experiments on the COCO dataset. Please see answer 1. above.
>
> 4. The models were trained on the original MNIST and ImageNet datasets. We trained the MNIST model ourselves, and retrieved the ImageNet models from [a] and [b]. All models were trained on single images and tested on grid-images illustrated in Figure 1. No L2 normalization was applied during training. Regarding multi-label training, we expect that our results also generalize to this case. This is because multi-label training would encourage the network to satisfy the delta-assumption, with delta-like responses in the feature maps corresponding to the objects present in the current image.
>
> 5. We have performed additional experiments with Dense Contrastive Learning [c], as suggested by the reviewer. The results are included in Appendix C in the revised manuscript.
>
> 6. In the experiments with L2 normalization, the normalization was only applied during _inference_ and not during training. The models were thus trained using a standard setup without L2 normalization. We have updated the paper to clarify this.
>
> 7. The tightness of the bound in Theorem 1 (Eq. (14)) depends entirely on the tightness of the bounds in Eqs. (5) (relaxed detector) and (8) (global pooling operator). This is illustrated by Corollary 1.2, where we show that when the inequalities in (5) and (8) become equalities, we also get equality in the result (Eq. (15)).
>
> 8. The delta assumption is not particularly strong in itself. It only assumes that there exists a certain type of images for which the output feature maps produce delta responses. However, whether the _object image types coincide with ground-truth classes_ in the dataset depends mostly on the classification capabilities of the model. Specifically, if a model is able to classify the dataset with high accuracy, it is more likely that the object image types coincide with the ground-truth classes.
>
> 9. Since our paper is centered on CNNs we have not included results with DINO, as it is primarily ViT-based. We have however, included additional results with Dense Contrastive Learning, which is another self-supervised model for images. See answer 5. above.
>
> [a] https://pytorch.org/vision/stable/models/resnet.html
>
> [b] https://lightning-bolts.readthedocs.io/en/0.3.4/self_supervised_models.html.
>
> [c] Dense Contrastive Learning for Self-Supervised Visual Pre-Training, CVPR-2021

---

> > ### Comment · Reviewer_zszo · 2023-11-07
> > **Thanks for the rebuttal**
> >
> > I apologize, my comments and recommendation were not visible to the authors. I am still on a border for this paper (but leaning towards an accept). The rebuttal helped address a few concerns but some still remain :
> >
> > I am glad the authors experimented with COCO as the additional dataset. But, an analysis of the slopes is not very insightful here. An actual counting task would have been a lot more interpretable. The paper still misses any experiments for the task of counting. The suggested benchmark (or others in object counting) would be useful here.
> >
> > Minor : Instead of just correlations, some more insights into the choice of SSL approach (SimCLR vs DenseCL) would be helpful given DenseCL seems to learn far better features for downstream tasks like detection, segmentation.

---

### Decision · Action_Editor_rbYa · 2023-11-09

**Recommendation:** Reject

**Comment:**

I, and the reviewers, believe that the paper presents an interest hypothesis that could be of interest to the community.

While some evidence is presented for the hypothesis, for the reasons outlined in "Claims", I believe that the paper does not quite reach the bar of "accurate, convincing and clear evidence". With further study, this could be a strong paper, but at this current stage I cannot recommend it for publication.

**Audience:**

The paper presents a very interesting hypothesis, and makes a rigorous attempt to support it with theory. I do believe there would be interest in this idea. Empirically, the study is somewhat limited by exploring only BatchNorm ResNets. To generate even more interest, empirical findings could be presented with more modern CNNs, or even Transformers (even if the theory doesn’t directly apply to those cases). Nonetheless, I think the idea could garner interest, or spur further study.

**Claims And Evidence:**

The paper presents some empirical evidence that "there is a monotonically increasing relationship between the norm of a representation, and the number of objects in the given image", in the specific setting of Batch-normalized ResNets. The reviewers found some of the evidence compelling. However, there were a few areas that left some doubt in the evidence:

1. The paper doesn't fully disentangle "object count" vs. "number of pixels covered by objects". The authors do provide a nice intuition, and empirical evidence on MNIST that object size is not the explaining factor. However, evidence on real data beyond MNIST would be nice to confirm this (e.g. using segmentation data, as suggested by the reviewer).

2. It appears that having multiple tiled images has a much stronger effect than multiple objects in an image. While statistically significant, the effect on real images (VOC and COCO) seems small. When using Dense Contrastive networks, the effect appears non-existent.

3. One source of evidence is improved classification accuracy after L2-normalization. This is quite compelling, however, the reviewers were not convinced by the line of reasoning that enforcing count-invariance should necessarily entail better accuracy. Further, in cases where accuracy and count are either weakly or inversely correlated, the accuracy still improves by a large margin, suggesting that another effect may be at play here.

4. No counting benchmark e.g. TallyQA or SOS (Zhang 2015) is presented, or comparison to numbers from the literature. It would be compelling to show that count can be predicted from norm on standard benchmarks, and performance compared to the literature.

**Resubmission Of Major Revision:**

The authors may consider submitting a major revision at a later time.